# GENOMIC NEXT-TOKEN PREDICTORS ARE IN-CONTEXT LEARNERS

## ABSTRACT

In-context learning (ICL) – the capacity of a model to infer and apply abstract patterns from examples provided within its input – has been extensively studied in large language models trained for next-token prediction on human text. In fact, prior work often attributes this emergent behavior to distinctive statistical properties in *human* language. This raises a fundamental question: can ICL arise *organically* in other sequence domains purely through large-scale predictive training?

To explore this, we turn to genomic sequences, an alternative symbolic domain rich in statistical structure. Specifically, we study the Evo2 genomic model, trained predominantly on next-nucleotide (A/T/C/G) prediction, at a scale comparable to mid-sized LLMs. We develop a controlled experimental framework comprising symbolic reasoning tasks instantiated in both linguistic and genomic forms, enabling direct comparison of ICL across genomic and linguistic models. Our results show that genomic models, like their linguistic counterparts, exhibit log-linear gains in pattern induction as the number of in-context demonstrations increases. To the best of our knowledge, this is the first evidence of organically emergent ICL in genomic sequences, supporting the hypothesis that ICL arises as a consequence of large-scale predictive modeling over rich data. These findings extend emergent meta-learning beyond language, pointing toward a unified, modality-agnostic view of in-context learning.

## 1 INTRODUCTION

Scaling Large Language Models (LLMs) has revealed an unexpected and powerful capacity: in-context learning (ICL) (Radford et al., 2018; Brown et al., 2020a), the ability to infer and apply abstract patterns purely from examples contained within their input. Almost all prominent evidence of emergent ICL so far (Srivastava et al., 2023, inter alia) comes from training on human language (e.g., English). This raises a fundamental question: *is there something inherently special about human language that enables ICL to emerge?* To investigate this, we turn to a radically different substrate: **the genome**. Genomic sequences can be viewed as a form of *natural* language that has evolved through nature. Like human language, they comprise complex symbol sequences that exhibit rich statistical regularities – motifs, repeats, dependencies (Benegas et al., 2025) that could, in principle, support pattern induction within context.

Testing for the existence of ICL requires large-scale models. We use the **Evo2** series (Brixi et al., 2025): a large-scale model trained solely on 'next-nucleotide' prediction. With training scale comparable to mid-sized LLMs (e.g., Qwen3-14B (Yang et al., 2025)), these genomic models finally make it possible to systematically compare genomic and linguistic representations under large-scale autoregressive training. If ICL arises primarily from scale and predictive compression, then large-scale genomic models such as Evo2 may already exhibit it.

Our experiment's results (§3) show that large genomic models do, in fact, exhibit ICL. **Both linguistic and genomic models exhibit log-linear improvements** in pattern inference as the number of demonstrations grows. We further validate these findings by demonstrating that Evo2 also exhibits robust few-shot learning on a real-world genomic promoter classification task (§3.1). These findings suggest *ICL is not an artifact that is specific to human language, but a broader consequence of*

---

LLMs were lightly used during the conception, implementation, and refinement of this paper. We assume full responsibility for all content.

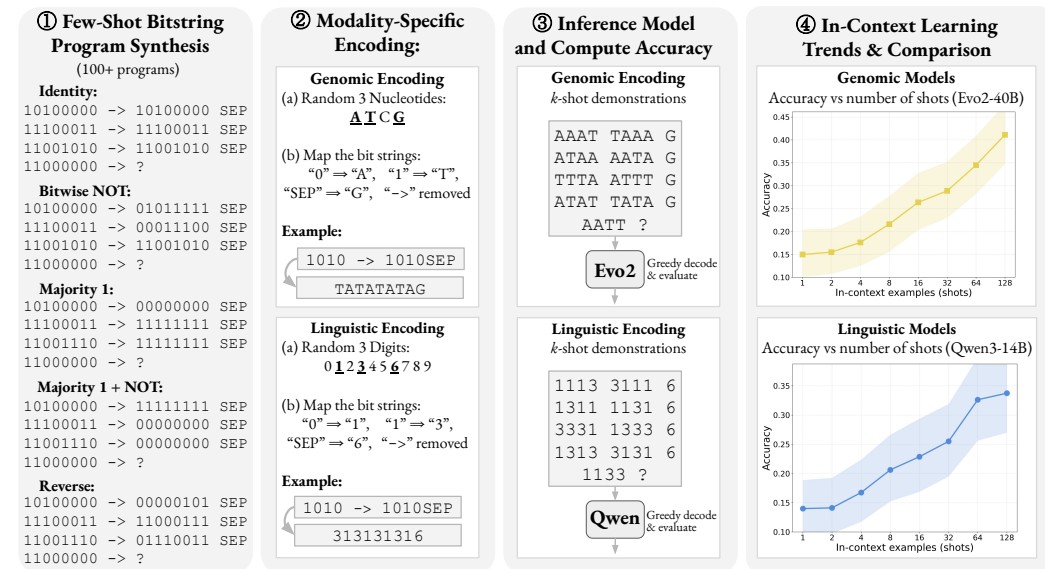

Figure 1: We design parallel symbolic reasoning tasks that allow direct comparison of ICL behavior across modalities. ① Few-shot bitstring program-synthesis tasks require models to infer mappings from examples. ② Each task is rendered in two modality-specific encodings. ③ Both genomic (Evo2) and linguistic (Qwen3) models receive $k$-shot demonstrations and are greedily decoded to compute exact-match accuracy. ④ **Both models show log-linear accuracy gains with more demonstrations**.

*compression and predictive modeling in pattern-rich sequence spaces*. This points toward a unified view of context-adaptive computation that spans different modalities of data.

## 2 FRAMEWORK FOR CROSS-DOMAIN IN-CONTEXT LEARNING

This section presents our framework for evaluating ICL in linguistic and genomic models. We outline experimental desiderata (§B), setup (§2.1), models (§C), and evaluation protocol (§2.2).

### 2.1 EXPERIMENTAL SETUP

**Notation:** We define in-context learning (ICL) as follows. Let $S$ and $O$ be the input and output domains, and let each task be a deterministic latent function $f : S \rightarrow O$. Given an ordered $n$-shot context $E = ((x_1, f(x_1)), \ldots, (x_n, f(x_n)))$ with $x_i \in S$ and a query $x \in S \setminus \{x_1, \ldots, x_n\}$, the model predicts $\hat{y} = M(E, x)$. A trial succeeds if $\hat{y} = f(x)$, i.e., $\mathbb{1}[\hat{y} = f(x)]$.

**Model families and selection rationale.** Details on the model families and our selection criteria are provided in Appendix C.

**Task Formulation: in-context program induction over abstract symbols.** Given our desiderata (§B), we construct symbolic induction tasks with a small lexicon: the model infers a hidden transformation from a few input–output demonstrations and applies it to a new query. Related setups appear in work on compositional reasoning (Brown et al., 2020b), Raven-style analogical reasoning (Raven et al., 1962; Webb et al., 2023), and ARC-AGI (Chollet, 2019).

**Bitstring domain and function space.** We use a shared symbolic domain $S = \{0, 1\}^k$ to ensure comparable token granularity across genomic and linguistic models. For genomic models, two nucleotides (e.g., A/C) encode bits while the remaining bases (G/T) serve as separators; for linguistic models, binary digits are typically single tokens (§G; Fig. 1). We set $k = 8$, so $|S| = 2^8 = 256$. Our task set is $F \subseteq \{f : S \rightarrow S\}$ with $|F| = 100$, where each $f$ is either one primitive transformation or a composition of two primitives.

**Generation of $F$.** We build $F$ from a library of 30 fundamental primitives (listed in §E.1) covering low-level logical operations and higher-order compositional structure. We construct the final function set, we use GPT-5-Codex to automatically generate 100 unique transformations by composing primitives into valid Python programs. The full list of transformations and corresponding source code is provided in §E.2.

## 2.2 EVALUATION PROTOCOL

**Evaluation metric.** For each transformation $f \in F$, sample i.i.d. trials $(E^{(t)}, x^{(t)})$ with $E^{(t)} \sim$ Unif$(E_n)$ and $x^{(t)} \sim$ Unif$(S \setminus E^{(t)})$, where $E_n = \{E \subset S : |E| = n\}$. Given $E^{(t)}$ and $x^{(t)}$, the model predicts $\hat{y}^{(t)} = M(E^{(t)}, x^{(t)})$ and a trial is correct if $\hat{y}^{(t)} = f(x^{(t)})$. Our Monte Carlo estimate of $n$-shot accuracy is

$$\hat{P}(M, n) = \frac{1}{|F|m} \sum_{f \in F} \sum_{t=1}^{m} \mathbb{1}\Big[M(E^{(t)}, x^{(t)}) = f(x^{(t)})\Big]. \tag{1}$$

**Model suites and sampling.** We use $m = 8$ Monte Carlo trials and shot counts $\mathcal{N} = \{1, 2, 4, 8, 16, 32, 64, 128\}$. Genomic models: $\mathcal{M}_G = \{\texttt{evo2\_1b\_base}, \texttt{evo2\_7b}, \texttt{evo2\_40b}\}$; linguistic models: $\mathcal{M}_L = \{\texttt{Qwen3-0.6B-Base}, \texttt{Qwen3-1.7B-Base}, \texttt{Qwen3-4B-Base}, \texttt{Qwen3-8B-Base}, \texttt{Qwen3-14B-Base}\}$. For each $(M, n) \in (\mathcal{M}_L \cup \mathcal{M}_G) \times \mathcal{N}$, we report $\hat{P}(M, n)$. Standard errors are estimated by a two-stage cluster bootstrap (resample $f$, then its trials) with 5000 replicates per $(M, n)$.

**Mode baseline.** We define a mode baseline that always predicts the most frequent output observed in the context. We formally define this baseline in §H.

## 3 EMPIRICAL RESULTS

This section reports empirical findings on few-shot bitstring generalization. §3.1 presents the main accuracy trends with respect to model size and shot count. §D.1 examines sensitivity to task complexity using a BitLoad measure, and §D.2 contrasts the models' qualitative competencies across individual transformations.

## 3.1 MAIN RESULTS

**Across both model families, accuracy generally rises linearly with respect to $\log(\textbf{shots})$.** A linear regression linking performance to $\log(\text{shots})$ yields highly significant slopes for all models (all $p \leq 10^{-3}$ via one-sided t-test on slope). As Fig. 3 shows, Evo2 models show cleanly monotonic gains, with a pronounced step from 1B to 7B, and near-indistinguishable curves for 7B and 40B; by 128 shots, both 7B and 40B surpass 40% accuracy (Fig. 3a). Qwen3 also trends upward overall but with non-monotonic patches—especially for smaller models in the 4–16 shot band—before resuming clear improvements from 32 to 128 shots; at 128 shots, the 14B model approaches 35% (Fig. 3b). A full table of all accuracies is in §I.

**The mode baseline's performance doesn't improve with more shots, and lags Qwen3 and Evo2 at high shot counts** (Fig. 3d). For Qwen3, advantages over the mode baseline become consistently significant ($p < 0.05$

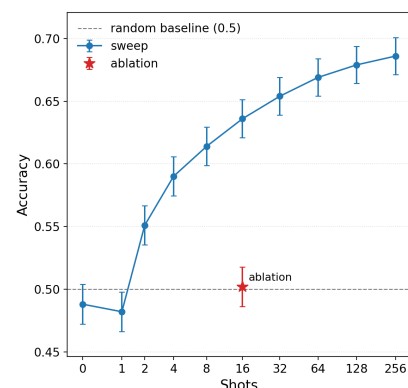

Figure 2: ICL Performance of Evo2 7B on `human_nontata_promoters` at varying shot counts. Error bars are $\pm$ one standard error. The performance shows clear monotonic improvements with respect to shot count.

via one-sided z-test on bootstrapped standard errors) at $n$=128 for all sizes greater than 0.6B. For Evo2, statistically significant advantages emerge slightly earlier: for 1B at $n$=64, the 7B at $n = 32$, and the 40B at $n = 16$. Regardless, all models surpass the naive baseline at high shot couts.

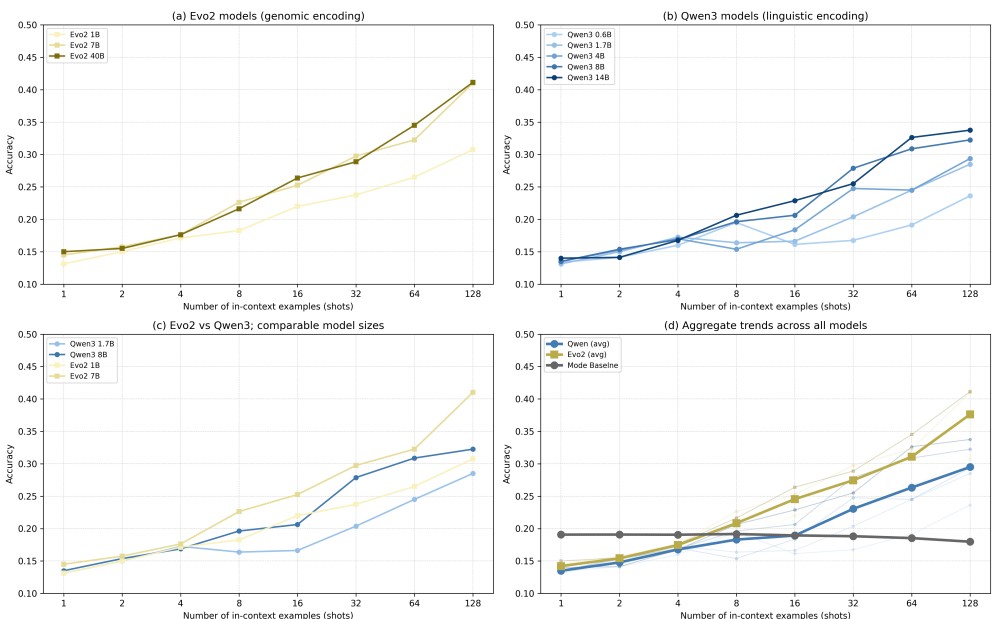

Figure 3: Few-shot performance of Qwen3 and Evo2 models. (a) Evo2 model performance with respect to log(shots). All models monotonically improve. 7B and 40B beat 1B. (b) Qwen3 model performance with respect to log(shots). All models improve, but not always monotonically. (c) At comparable sizes, Evo2 outperforms Qwen3. (d) Averaged performance across both model families. All models exceed the mode baseline shown in gray color. The exact accuracies and error bars (bootstrap-based standard errors) are included in §I.

.

**Evo2 performs ICL on genomic tasks.** In Fig. 2, we demonstrate that Evo2 7B can also perform few-shot in-context learning (ICL) on a relatively in-distribution binary genomic classification task: `human_nontata_promoters`, in which the model must predict whether a 251 nucleotide-long human DNA sequence is a promoter. Details of this experiment can be found in §K.

**Further Analysis.** We provide additional analyses of *BitLoad* (how many input bits influence the output) sensitivity in §D.1, *BitDiversity* (the number of minority bits in the output string) sensitivity in §L, and qualitative task differences in §D.2. Finally, we also provide analysis of how model scale impacts accuracy in §J.

## 4 CONCLUSION

We introduce a suite of bitstring reasoning tasks that can be encoded in both natural language and genomic sequences, showing that genomic models – like their linguistic counterparts – exhibit clear in-context learning. Across all Evo2 model sizes, we observe robust log-linear gains in accuracy with increasing demonstrations, paralleling the scaling trends of Qwen3 language models. These findings challenge the notion that ICL is unique to human language, suggesting it emerges whenever an expressive model is trained autoregressively on structured, pattern-rich data. See §A for a broader discussion of implications.

**Potential future work:** This work motivates searching for ICL in other non-linguistic modalities – time series (Das et al., 2024), system logs (Akhauri et al., 2025), physics simulations (Holzschuh et al., 2025), chess games (Ruoss et al., 2024), and climate projections (Duncan et al., 2025). Each offers a structured, patterned substrate that could support its own form of contextual reasoning. These diverse modalities, each with their unique structure and constraints, suggest a rich world of non-linguistic ICL capability waiting to be explored, and this work represents a maiden voyage into these extremely interesting waters.

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

# A    DISCUSSION

**What are the implications of our findings on prior efforts to explain the emergence of ICL?**
First, let us organize the existing frameworks for pinpointing the conditions under which ICL
emerges:

(E1) **ICL's emergence is due to data distributional properties:** The distributional properties of
data, such as "parallel structures" in human language pretraining data (Chen et al., 2024),
its compositional structure (Hahn & Goyal, 2023), "burstiness" (Chan et al., 2022) and other
such properties (Wibisono & Wang, 2024; Reddy, 2023) may be of importance (and perhaps
necessary) for the emergence of ICL.

(E2) **ICL's emergence is due to a compression mechanism:** The large-scale compression mech-
anism during massive pretraining might drive ICL (Elmoznino et al., 2024a;b; Hahn & Goyal,
2023).

(E3) **ICL's emergence may require specific architectural properties:** While Transformers might
be better suited for ICL than LSTMs (Xie et al., 2021), evidence is mixed (Lee et al., 2023),
and non-Transformer models have also demonstrated ICL capabilities (Grazzi et al., 2024;
Park et al., 2024).

Our findings refine existing hypotheses about ICL's origins. The emergence of ICL in genomic
models challenges accounts that rely solely on language-specific distributional structures (E1). The
presence of ICL across both genomic and linguistic models supports the compression-based expla-
nation (E2), suggesting that large-scale sequence compression and its induced inductive biases drive
ICL across modalities. With respect to architecture (E3), Evo2 – an autoregressive hybrid combining
convolutional and attention layers rather than a pure Transformer – exhibits similar scaling behavior,
indicating that ICL does not depend on the pure Transformer form. Instead, architecture provides an
expressive substrate that enables pattern induction once exposed to sufficiently large and structured
data. Overall, these results position ICL as a modality-agnostic outcome of large-scale next-token
prediction, rather than a phenomenon tied to linguistic statistics or a specific architecture.

**What are the implications of our findings on the frameworks to explain how ICL operates?**
We next consider the major perspectives that seek to explain how ICL operates. One view holds that
ICL functions as a mix of *task learning* and *task retrieval*, with demonstrations serving either to re-
call pretrained capabilities or to enable learning on the fly (Pan et al., 2023; Lin & Lee, 2024; Wang
et al., 2024; Fang et al., 2025). Our symbolic reasoning tasks, instantiated in both linguistic and
genomic domains, provide direct evidence for this *task learning* mode, aligning with this hypothesis
and prior work (Pan et al., 2023; Fang et al., 2025). Because these tasks do not depend on pre-
trained semantic priors, they do *not* invoke task *retrieval*, offering limited insight into the Bayesian
view that interprets ICL as implicit inference over latent concepts (Xie et al., 2021; Panwar et al.,
2023; Wang et al., 2023b; Jiang et al., 2024). Meanwhile, our results remain agnostic toward the
optimization-based hypothesis, which posits that ICL implements an implicit gradient-descent-like
process (Akyürek et al., 2022; Ahn et al., 2023; Mahankali et al., 2023; Li et al., 2023), as well as
the induction-based account, which attributes ICL to specialized "circuits" for performing inductive
generalization (Elhage et al., 2021; Olsson et al., 2022; Wang et al., 2023a; Bansal et al., 2023;
Ren et al., 2024). Together, our results most strongly support the presence of genuine task *learning*
within ICL.

**Does Evo2 have an innate advantage on these tasks?** Possibly, for multiple reasons. First, though
Evo2's is trained on less tokens total, all of Evo2's training tokens are long sequences of repeated
nucleotides, and very little of Qwen3's training tokens are long sequences of repeated digits. Second,
Evo2's StripedHyena2 architecture was found to to significantly outperform a vanilla transformer
on long DNA sequences (Brixi et al., 2025). This could give Evo2 an innate advantage on long
contexts containing the same few symbols vs Qwen3. Our result results do not imply Evo2's ICL
ability is superior to Qwen3's, and we concede that our few-shot prompting setup may be biased
toward Evo2. Attempts to make the task more legible to Qwen3, however, ran into the confound
that Qwen3 has pretraining exposure to bitstring manipulation. Any prompting setup that included
0s and 1s immediately resulted in an extreme increase in Qwen3's performance. Future work is
needed to establish a method that controls for Evo2's structural advantages without granting Qwen3
an unfair edge via its pretraining knowledge.

**Why not test the models on semantic tasks?** Semantic tasks – such as identifying the capitals associated with countries, classifying malformed proteins, etc. – require a fair amount of pretraining exposure to the concepts involved in the task as well as measuring ICL. While it's undoubtedly ICL when a model infers a semantic transformation (for instance, country→capital, word→opposite), the pretraining knowledge necessary to manifest this ICL precludes its use in extreme cross-modality comparisons. Focusing on far simpler bitstring transformations that can be learned entirely in-context allows for an apples-to-apples comparison between Evo2 and Qwen3. To ensure that the ICL observed is not an artifact of this simplified domain, we additionally demonstrate in Appendix K that Evo2 exhibits robust, scaling ICL on a native genomic task (promoter classification), though we exclude this from the main comparison to maintain parity with the linguistic models.

## B  Experimental Desiderata

**Desideratum 1: Cross-domain comparability.** Our experimental framework requires that each task be performable by *both* language and genomic models. Thus, every task must be representable in both linguistic and nucleotide alphabets. This constraint excludes existing benchmarks that rely on domain-specific semantics – such as language reasoning datasets (e.g., BIG-Bench, StrategyQA, GSM8K (Srivastava et al., 2023; Geva et al., 2021; Cobbe et al., 2021)) or biological tasks (e.g. variant effect prediction, exon identification (Brixi et al., 2025)). While one could, in principle, translate domain-specific tasks into an alternate alphabet (e.g., mapping language tokens to base sequences via quaternary encoding), doing so inherently biases the evaluation: the source domain retains advantages, while the target domain must operate on representations that are unnatural to it. As a result, such cross-domain encodings confound the comparison, reflecting representational translation artifacts rather than genuine differences in ICL behavior.

**Desideratum 2: Limited vocabulary.** Since genomic models operate on only four nucleotides (A, T, C, G), tasks must be expressible within an equally compact alphabet. This rules out the existing symbolic reasoning and analogy benchmarks (Hodel & West, 2023; Lewis & Mitchell, 2024; Webb et al., 2025), which rely on richer vocabularies (e.g., shapes, colors, or linguistic tokens with explicit semantic roles). Such tasks cannot be faithfully represented in a four-token regime without introducing artifacts or structural loss. Accordingly, our evaluation focuses on tasks that retain abstract reasoning structure while remaining compatible with the low-vocabulary symbolic space shared across linguistic and genomic models.

## C  Model Families and Selection Rationale

For fair cross-domain comparison, we use two representative model families: Qwen3 for human language and Evo2 for genomics. The rationale for this selection is as follows:

(a) *Parameter scaling:* Both families span multiple orders of magnitude in parameter count, enabling systematic scaling analysis of ICL ability. The Qwen3 series ranges from 0.6B to 14B parameters, while Evo2 includes 1B, 7B, and 40B models (Yang et al., 2025; Brixi et al., 2025). This parallel scaling structure facilitates consistent measurement of how ICL performance evolves with model size across linguistic and genomic modalities.

(b) *Compute matching:* The largest models in each family are trained with comparable total compute, offering an opportunity for an approximately compute-matched cross-domain comparison. Using the standard $6ND$ estimate (Kaplan et al., 2020), Qwen3-14B-Base is trained with about $3.2 \times 10^{24}$ FLOPs, while Evo2-40B is trained with $2.25 \times 10^{24}$ FLOPs (Yang et al., 2025; Brixi et al., 2025), making Evo2 uniquely well-suited for comparison with Qwen3 at scale.

(c) *Availability of base models:* The Qwen3 family releases base (*pre*-instruction-tuned) models at all scales up to 14B parameters, enabling direct evaluation of the intrinsic inductive reasoning ability of pure next-token predictors, without instruction-tuning artifacts. The Evo2 base models have no additional instruction tuning applied—the only nuance is that Evo2 1B is trained at a context length of 8192 nucleotides, whereas the 7B/40B are extended to a context length of one million.

(d) *Tokenizer:* Qwen3 uses a standard BPE tokenizer with a vocabulary size of 151,669. Evo2 uses a byte-level tokenizer, so individual nucleotides are mapped to individual tokens. Notably, Qwen's tokenizer maps single digits to single tokens, allowing for parity in our experiments (Yang et al., 2025; Brixi et al., 2025).

(e) *Context length:* Qwen3's 0.6B and 1.7B models have a context length of 32K. The remaining dense models have a context length of 128K. Evo2's 1B model has a context length of 8K, and the remaining models have a context length of 1M. As none of our experiments approach these limits, we can use all models in both families without fear of context length as a confound (Yang et al., 2025; Brixi et al., 2025).

(f) *Training corpora:* All Qwen3 models are trained on 36 trillion text tokens covering a wide variety of topics and languages. Evo1 1B is trained on 1 trillion tokens, Evo2 7B is trained on 2.4 trillion tokens, and Evo2 40B is trained on 9.3 trillion tokens. These extensive training corpora ensure that any potential ICL dynamics have thoroughly emerged (Yang et al., 2025; Brixi et al., 2025).

(g) *Architecture:* Qwen3 is based on a conventional Llama-like transformer architecture, while Evo2 uses the StripedHyena2 architecture which intersperses convolutional layers with attention-based ones. While ideally both model families would use the same architecture, there were no vanilla transformers at Evo2's compute scale (Yang et al., 2025; Brixi et al., 2025).

(h) *Licensing:* All models are released under Apache 2.0 (Yang et al., 2025; Arc Institute, 2025a;c;b), ensuring reproducibility.

# D  SUPPLEMENTARY ANALYSES OF TASK COMPLEXITY AND QUALITATIVE DIFFERENCES

## D.1  ICL SENSITIVITY TO TASK COMPLEXITY: BITLOAD ANALYSIS

To understand which transformations Qwen3 and Evo2 infer most effectively, we analyze their performance across varying task complexities. We focus on the largest models in each family (Qwen3-14B and Evo2-40B) under the ($n = 128$) shot regime, providing both models ample opportunity to display their ICL abilities.

**Defining BitLoad.** We introduce *BitLoad*, a measure of a function's intrinsic complexity. Informally, BitLoad quantifies how many input bits influence the output. Formally, it is defined as:

$$\text{BitLoad}(f) = \sum_{i=1}^{k} \mathbb{1}\Big[\exists x, j : f_j(x) \neq f_j(x^{\oplus i})\Big],$$

(2)

where $x^{\oplus i}$ denotes $x$ with bit $i$ flipped, and $f_j(x)$ returns the $j$-th output bit. Intuitively, it counts the number of bit positions whose perturbation changes the output. So, the larger the BitLoad of a function, the harder it is since it requires the model to attend to more bits. This metric is similar to the existing statistical measure of "relevant features", just defined specifically for our bitstring manipulation tasks. (Nevo & El-Yaniv, 2002).

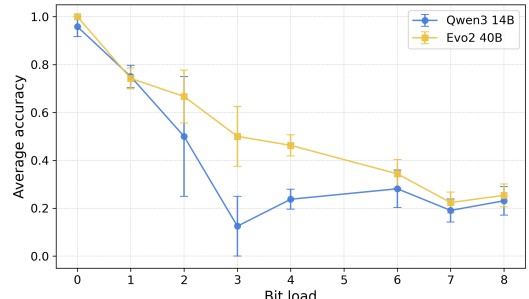

Figure 4: Accuracy vs. BitLoad averaged across all tasks (BitLoad; Eq.2). Qwen declines sharply with increasing BitLoad, while Evo degrades more gradually, indicating greater robustness. Details in §D.1.

Fig. 4 shows the mean accuracy of Qwen vs Evo with respect to the BitLoad of all of our tasks. A full table of the BitLoad of every tested task is attached in the appendix E.3. We see that both models achieve near-perfect accuracy on constant (0 BitLoad) tasks and remain similar at BitLoad 1. However, by BitLoad 2, Evo begins to outperform Qwen, and beyond that point Qwen's accuracy drops sharply – falling below 20% by BitLoad 4 – while Evo's performance declines more gradually, remaining above 40% before converging near 20% at BitLoad 8. This asymmetry suggests that Evo

maintains partial generalization as dependency depth increases (BitLoad value), whereas Qwen's ICL collapses more abruptly.

While BitLoad strongly correlates with task accuracy, Fig. 4 also shows notable deviations within the error bars. Thus, while task complexity is a strong predictor of ICL accuracy, other factors (e.g., transformation depth, pretraining exposure) likely play meaningful roles, which motivate our qualitative study in §D.2.

### D.2   QUALITATIVE ANALYSIS OF FUNCTIONAL AND BEHAVIORAL DIFFERENCES

To complement the quantitative BitLoad analysis (§D.1), we conduct a per-task comparison to identify qualitative differences between Qwen3 and Evo2. Specifically, we analyze which tasks each model performs best on and how their inductive profiles diverge. We focus on the $n = 128$ shot regime for the largest models – Qwen3-14B and Evo2-40B – where both have maximal opportunity to exhibit ICL. We rank all tasks by model accuracy and examine the top ten for each model. Tasks that appear in one model's top ten but not the other are considered "exclusive."

**Exclusive competencies.** Qwen's exclusive tasks involve right-shift operations: `"spread_last_bit"` $\rightarrow$ `"shift_right_zero"` and `"edge_mask"` $\rightarrow$ `"shift_right_zero"`. For instance, "shift_right_zero" pads the bitstring on the left with a zero and truncates the last bit (e.g., 01010000 $\rightarrow$ 00101000). Qwen achieves 100% accuracy on `"spread_last_bit"` $\rightarrow$ `"shift_right_zero"`, whereas Evo2 achieves only 50%. A similar but smaller gap appears for `"edge_mask"` $\rightarrow$ `"shift_right_zero"` (87.5% vs. 62.5%).

In contrast, Evo2's exclusive strengths involve multi-bit transformations. A clear example is `"flip_bits"` $\rightarrow$ `"right_half"`, which applies bitwise NOT followed by masking the first half of the input (e.g., 01011100 $\rightarrow$ 00000011). Evo2 achieves 87.5% accuracy, while Qwen only 25%. This task has a BitLoad of four, consistent with Evo's superior performance on medium-complexity (2–4 bit) transformations observed in Fig. 4.

**Shared strengths.** Despite these differences, 7 of the top 10 tasks are shared between Qwen and Evo, yielding an intersection-over-union of 0.54. Both models excel at simple transformations such as constant-outputs or single-bit dependencies, e.g., `"spread_last_bit"` which copies the final bit to all positions.

**Differential skill profiles.** To sharpen contrasts, we identify the ten tasks most favoring each model. Qwen's advantages are concentrated in simple shifts and aggregation tasks. It outperforms Evo by 37.5% on the `"minority"` operation (output all 1s if zeros > ones), and by a similar margin on two parity-based tasks requiring counting the number of 1s. These trends suggest that Qwen may be better at reasoning over global properties of bitstrings. Its superiority on simple shifts may also be explained by the extreme rarity of frame shift mutations in DNA due to how catastrophic they are – a single nucleotide offset can decimate an entire protein. This is empirically supported by the fact that single-nucleotide deletions/shifts increase perplexity far more than other common mutations when presented to Evo2 (Brixi et al., 2025).

Evo2, by contrast, dominates tasks requiring full-bitstring manipulation. It achieves 62.5% on bitwise NOT (vs. 0% for Qwen), 62.5% on identity (vs. 12.5%), and large margins on compositions such as `"flip_bits"` $\rightarrow$ `"right_half"` (87.5% vs. 25%) and `"rotl1"` $\rightarrow$ `"flip_bits"` (37.5% vs. 0%). This exposes perhaps the most important difference between Qwen and Evo's ICL in this specific context: Evo can learn simple full-bitstring operations in-context, whereas Qwen cannot. Notably, Qwen3's base models are trivially capable of learning the identity in a more familiar few-shot context – when examples are presented with arrows and newlines separating them, instead of our intentionally unfamiliar encoding. Thus these results should be taken as an existence proof of ICL in Evo2, not a definitive statement of Evo2 having more ICL ability than Qwen. We leave a comparison of these models across broader tasks to future work.

# E   Further Details on Synthetic Task Definition

## E.1   Table of Primitives

The following table describes the thirty primitives used to construct the task space via composition – their use is described in §2.1.

| Primitive | Description |
|---|---|
| alternating_start_one | Produce a mask marking positions that differ from the alternating 1010... pattern starting with 1 (1 = mismatch, 0 = match). |
| alternating_start_zero | Produce a mask marking positions that differ from the alternating 0101... pattern starting with 0 (1 = mismatch, 0 = match). |
| center_mask | Zero out the first and last bits while leaving the interior bits unchanged; strings of length $\leq 2$ become all zeros. |
| double_rotl | Circularly rotate the bitstring two positions to the left. |
| double_rotr | Circularly rotate the bitstring two positions to the right. |
| edge_mask | Preserve the first and last bits and zero out every interior bit (length 0/1 strings pass through). |
| flip_bits | Invert every bit, swapping 0s for 1s and vice versa. |
| identity | Return the bitstring unchanged. |
| invert_prefix | Flip the bits in the left half of the string, keep the right half as is. |
| invert_suffix | Keep the left half as is and flip every bit in the right half of the string. |
| keep_even_positions | Keep bits at even indices (0-based) and zero out bits at odd indices. |
| keep_odd_positions | Keep bits at odd indices (0-based) and zero out bits at even indices. |
| left_half | Preserve the left half of the string and replace the right half with zeros. |
| majority | Fill the string with the majority bit from the input; ties resolve to all 1s. |
| meta_constant | Returns a random, pre-set constant. |
| minority | Fill the string with the minority bit from the input; ties resolve to all 0s. |
| mirror_half | Copy the left half of the string onto the right half in reverse order, keeping the center bit unchanged for odd lengths. |
| ones_if_palindrome | Output all 1s if the input is a palindrome; otherwise output all 0s. |
| parity_fill | Output all 1s when the input contains an odd number of 1s; otherwise output all 0s. |
| reverse_bits | Reverse the order of the bits in the string. |
| right_half | Zero out the left half and keep the right half unchanged. |
| rotl1 | Circularly rotate the bitstring one position to the left. |
| rotr1 | Circularly rotate the bitstring one position to the right. |
| shift_left_zero | Shift the string left by one, dropping the first bit and appending a 0 on the right. |
| shift_right_zero | Shift the string right by one, inserting a 0 on the left and dropping the last bit. |
| spread_first_bit | Replace every position with the first bit of the input. |
| spread_last_bit | Replace every position with the last bit of the input. |
| swap_halves | Swap the left and right halves of the string. |
| swap_pairs | Swap each adjacent pair of bits (positions 0/1, 2/3, ...). |
| xor_with_s0 | Can only be applied after another primitive. Computes the logical XOR between the original input $s_0$ and the output of the first primitive. |

Table 1: The 30 unary primitives used to construct functions in $F$.

## E.2 TABLE OF FUNCTIONS

The following table describes the one hundred specific compositions of primitives used to construct the evaluation suite in §**??**.

| Function | Function | Function |
|---|---|---|
| identity | spread_last_bit | swap_halves → shift_left_zero |
| rotl1 | invert_prefix | swap_halves → shift_right_zero |
| reverse_bits | invert_suffix | shift_left_zero → swap_halves |
| flip_bits | meta_constant | shift_right_zero → swap_halves |
| swap_halves | flip_bits → reverse_bits | keep_even_positions → flip_bits |
| majority | rotl1 → reverse_bits | keep_odd_positions → flip_bits |
| minority | reverse_bits → rotl1 | flip_bits → keep_even_positions |
| parity_fill | rotl1 → flip_bits | flip_bits → keep_odd_positions |
| alternating_start_one | swap_halves → reverse_bits | edge_mask → flip_bits |
| alternating_start_zero | swap_halves → flip_bits | center_mask → flip_bits |
| left_half | double_rotl → flip_bits | shift_left_zero → keep_even_positions |
| right_half | rotr1 → flip_bits | shift_left_zero → keep_odd_positions |
| double_rotl | spread_first_bit → flip_bits | shift_right_zero → keep_even_positions |
| rotr1 | spread_last_bit → flip_bits | shift_right_zero → keep_odd_positions |
| double_rotr | left_half → flip_bits | keep_even_positions → reverse_bits |
| ones_if_palindrome | right_half → flip_bits | keep_odd_positions → reverse_bits |
| mirror_half | flip_bits → left_half | shift_left_zero → parity_fill |
| spread_first_bit | flip_bits → right_half | shift_right_zero → parity_fill |
| spread_last_bit | double_rotl → reverse_bits | parity_fill → shift_left_zero |
| invert_prefix | rotl1 → swap_halves | parity_fill → shift_right_zero |
| invert_suffix | xor_with_s0 | spread_first_bit → shift_left_zero |
| meta_constant | flip_bits → xor_with_s0 | spread_last_bit → shift_right_zero |
| shift_left_zero | ones_if_palindrome → flip_bits | spread_first_bit → keep_even_positions |
| shift_right_zero | flip_bits → mirror_half | spread_last_bit → keep_odd_positions |
| swap_pairs | invert_prefix → reverse_bits | spread_first_bit → edge_mask |
| keep_even_positions | left_half → reverse_bits | spread_last_bit → edge_mask |
| keep_odd_positions | right_half → reverse_bits | spread_first_bit → center_mask |
| edge_mask | parity_fill → flip_bits | spread_last_bit → center_mask |
| center_mask | rotl1 → spread_first_bit | rotl1 → shift_left_zero |
| xor_with_s0 | shift_left_zero → flip_bits | rotl1 → shift_right_zero |
| flip_bits → reverse_bits | shift_right_zero → flip_bits | shift_left_zero → rotl1 |
| rotl1 → reverse_bits | flip_bits → shift_left_zero | shift_right_zero → rotl1 |
| reverse_bits → rotl1 | flip_bits → shift_right_zero | reverse_bits → edge_mask |
| rotl1 → flip_bits | swap_pairs → flip_bits | reverse_bits → center_mask |
| swap_halves → reverse_bits | shift_left_zero → reverse_bits | edge_mask → shift_left_zero |
| swap_halves → flip_bits | shift_right_zero → reverse_bits | edge_mask → shift_right_zero |
| double_rotl → flip_bits | spread_first_bit → flip_bits | shift_left_zero → shift_left_zero |
| rotr1 → flip_bits | shift_left_zero → edge_mask | shift_left_zero → swap_pairs |

Table 2: The complete set of 100 functions in $F$, consisting of 30 single primitives and 70 composed functions $((f \rightarrow g)(x) = g(f(x)))$.

## E.3 BitLoad of Functions

| Function / Composition | BitLoad | Function / Composition | BitLoad |
|---|---|---|---|
| identity | 8 | rotl1 | 8 |
| reverse_bits | 8 | flip_bits | 8 |
| swap_halves | 8 | majority | 8 |
| minority | 8 | parity_fill | 8 |
| alternating_start_one | 8 | alternating_start_zero | 8 |
| left_half | 4 | right_half | 4 |
| double_rotl | 8 | rotr1 | 8 |
| double_rotr | 8 | ones_if_palindrome | 8 |
| mirror_half | 4 | spread_first_bit | 1 |
| spread_last_bit | 1 | invert_prefix | 8 |
| invert_suffix | 8 | meta_constant | 0 |
| flip_bits → reverse_bits | 8 | rotl1 → reverse_bits | 8 |
| reverse_bits → rotl1 | 8 | rotl1 → flip_bits | 8 |
| swap_halves → reverse_bits | 8 | swap_halves → flip_bits | 8 |
| double_rotl → flip_bits | 8 | rotr1 → flip_bits | 8 |
| spread_first_bit → flip_bits | 1 | spread_last_bit → flip_bits | 1 |
| left_half → flip_bits | 4 | right_half → flip_bits | 4 |
| flip_bits → left_half | 4 | flip_bits → right_half | 4 |
| double_rotl → reverse_bits | 8 | rotl1 → swap_halves | 8 |
| xor_with_s0 | 0 | flip_bits → xor_with_s0 | 0 |
| ones_if_palindrome → flip_bits | 8 | flip_bits → mirror_half | 4 |
| invert_prefix → reverse_bits | 8 | left_half → reverse_bits | 4 |
| right_half → reverse_bits | 4 | parity_fill → flip_bits | 8 |
| rotl1 → spread_first_bit | 1 | shift_left_zero | 7 |
| shift_right_zero | 7 | swap_pairs | 8 |
| keep_even_positions | 4 | keep_odd_positions | 4 |
| edge_mask | 2 | center_mask | 6 |
| shift_left_zero → flip_bits | 7 | shift_right_zero → flip_bits | 7 |
| flip_bits → shift_left_zero | 7 | flip_bits → shift_right_zero | 7 |
| swap_pairs → flip_bits | 8 | shift_left_zero → reverse_bits | 7 |
| shift_right_zero → reverse_bits | 7 | swap_halves → shift_left_zero | 7 |
| swap_halves → shift_right_zero | 7 | shift_left_zero → swap_halves | 7 |
| shift_right_zero → swap_halves | 7 | keep_even_positions → flip_bits | 4 |
| keep_odd_positions → flip_bits | 4 | flip_bits → keep_even_positions | 4 |
| flip_bits → keep_odd_positions | 4 | edge_mask → flip_bits | 2 |
| center_mask → flip_bits | 6 | shift_left_zero → keep_even_positions | 4 |
| shift_left_zero → keep_odd_positions | 3 | shift_right_zero → keep_even_positions | 3 |
| shift_right_zero → keep_odd_positions | 4 | keep_even_positions → reverse_bits | 4 |
| keep_odd_positions → reverse_bits | 4 | shift_left_zero → parity_fill | 7 |
| shift_right_zero → parity_fill | 7 | parity_fill → shift_left_zero | 8 |
| parity_fill → shift_right_zero | 8 | spread_first_bit → shift_left_zero | 1 |
| spread_last_bit → shift_right_zero | 1 | spread_first_bit → keep_even_positions | 1 |
| spread_last_bit → keep_odd_positions | 1 | spread_first_bit → edge_mask | 1 |
| spread_last_bit → edge_mask | 1 | spread_first_bit → center_mask | 1 |
| spread_last_bit → center_mask | 1 | rotl1 → shift_left_zero | 7 |
| rotl1 → shift_right_zero | 7 | shift_left_zero → rotl1 | 7 |
| shift_right_zero → rotl1 | 7 | reverse_bits → edge_mask | 2 |
| reverse_bits → center_mask | 6 | edge_mask → shift_left_zero | 1 |
| edge_mask → shift_right_zero | 1 | shift_left_zero → shift_left_zero | 6 |
| shift_left_zero → swap_pairs | 7 | shift_left_zero → edge_mask | 1 |

Table 3: BitLoad for every primitive and composed function in $F$. Used in §D.1.

## F    A GLOBAL METRIC OVER ALL PROGRAMS

For a given function $f$, model $M$, and number of in-context examples $n$, we define the average accuracy over all context sets $E_N = \{E \subset S : |E| = n\}$:

$$A_f(M, n) = \frac{1}{|E_N|(|S| - n)} \sum_{E \in E_N} \sum_{x \in S \setminus E} A(f, E, x, M).$$

The overall benchmark score across all functions $F$ is then

$$P(M, n) = \frac{1}{|F|} \sum_{f \in F} A_f(M, n).$$

Because exact evaluation is intractable (for $n = 8$, $|E_N|(|S| - N) = \binom{|S|}{n}(|S| - n) = \binom{256}{8} \cdot 248 \approx 1.016 \times 10^{17}$) we estimate $A_f(M, n)$ via Monte Carlo, as discussed in §2.2.

## G    PROMPT ENCODING

Prompts are encoded using a unified symbolic scheme to enable evaluation across linguistic and genomic models. For linguistic models, bits $(0, 1)$ are mapped to two random digits $(0\text{–}9)$; for genomic models, to random nucleotides $(A, T, C, G)$. The separator token $(\rightarrow)$ is omitted, and in-context examples are separated by a randomly chosen unused token distinct from those representing 0 and 1. (See Fig. 1 for examples.) All mappings are randomized per trial to avoid memorization or positional bias.

## H    MODE BASELINE

We define a mode baseline that always predicts the most frequent output observed in the context. For a given function $f$, context set $E \subset S$, and query input $x \in S \setminus E$, the mode prediction is $\hat{y}_{\text{mode}}(f, E, x) = \arg\max_{y \in S} \left|\{e \in E : f(e) = y\}\right|$, where ties in the $\arg\max$ are broken randomly. This simply corresponds to guessing the most common output in the few-shot examples. The overall mode baseline with $n$ shots and across the set of all functions $F$ is:

$$\hat{P}_{\text{mode}}(n) = \frac{1}{m|F|} \sum_{f \in F} \sum_{t=1}^{m} \mathbb{1}\left[\hat{y}_{\text{mode}}(f, E^{(t)}, x^{(t)}) = f(x^{(t)})\right], \tag{3}$$

where $E^{(t)}$ and $x^{(t)}$ are sampled identically to the model evaluation in Eq.1. This baseline corresponds to making an educated guess based only on the overall distribution of function outputs that simply learns the majority statistics of the prompt without attempting to infer the underlying transformation.

## I    FULL ACCURACY RESULTS

Here we show the full table of accuracies used in §3.1 to analyze the ICL capabilities of Evo2 and Qwen3 and perform the necessary statistical tests.

| Model | 1 Shot | 2 Shots | 4 Shots | 8 Shots | 16 Shots | 32 Shots | 64 Shots | 128 Shots |
|---|---|---|---|---|---|---|---|---|
| Qwen3 0.6B | $13.4_{\pm2.3}$ | $14.1_{\pm2.3}$ | $16.0_{\pm2.6}$ | $19.5_{\pm2.7}$ | $16.1_{\pm2.4}$ | $16.8_{\pm2.5}$ | $19.1_{\pm2.7}$ | $23.6_{\pm2.9}$ |
| Qwen3 1.7B | $13.1_{\pm2.2}$ | $15.0_{\pm2.7}$ | $\mathbf{17.2}_{\pm2.8}$ | $16.4_{\pm2.6}$ | $16.6_{\pm2.7}$ | $20.4_{\pm2.9}$ | $24.5_{\pm3.4}$ | $28.5_{\pm3.5}$ |
| Qwen3 4B | $13.5_{\pm2.4}$ | $15.2_{\pm2.4}$ | $17.0_{\pm2.6}$ | $15.4_{\pm2.5}$ | $18.4_{\pm2.7}$ | $24.8_{\pm3.2}$ | $24.5_{\pm3.4}$ | $29.4_{\pm3.4}$ |
| Qwen3 8B | $13.5_{\pm2.3}$ | $\mathbf{15.4}_{\pm2.5}$ | $16.9_{\pm2.7}$ | $19.6_{\pm2.7}$ | $20.6_{\pm2.8}$ | $\mathbf{27.9}_{\pm3.3}$ | $30.9_{\pm3.5}$ | $32.2_{\pm3.5}$ |
| Qwen3 14B | $\mathbf{14.0}_{\pm2.4}$ | $14.1_{\pm2.4}$ | $16.8_{\pm2.7}$ | $\mathbf{20.6}_{\pm2.9}$ | $\mathbf{22.9}_{\pm3.1}$ | $25.5_{\pm3.2}$ | $\mathbf{32.6}_{\pm3.7}$ | $\mathbf{33.8}_{\pm3.5}$ |
| Evo2 1B | $13.1_{\pm2.2}$ | $15.0_{\pm2.4}$ | $17.1_{\pm2.9}$ | $18.2_{\pm2.7}$ | $22.0_{\pm2.9}$ | $23.8_{\pm2.8}$ | $26.5_{\pm3.0}$ | $30.8_{\pm3.1}$ |
| Evo2 7B | $14.5_{\pm2.4}$ | $\mathbf{15.8}_{\pm2.7}$ | $\mathbf{17.6}_{\pm2.9}$ | $\mathbf{22.6}_{\pm2.9}$ | $25.2_{\pm3.0}$ | $\mathbf{29.8}_{\pm3.1}$ | $32.2_{\pm3.1}$ | $41.0_{\pm3.4}$ |
| Evo2 40B | $\mathbf{15.0}_{\pm2.6}$ | $15.5_{\pm2.5}$ | $\mathbf{17.6}_{\pm2.8}$ | $21.6_{\pm3.1}$ | $\mathbf{26.4}_{\pm3.1}$ | $28.9_{\pm3.1}$ | $\mathbf{34.5}_{\pm3.2}$ | $\mathbf{41.1}_{\pm3.3}$ |

Table 4: In-context learning performance across model families and shot counts. Values show accuracy ± standard error. All numbers are percentages. **Bold** numbers show the best performance within a model family. Models are ordered by parameter count within each family.

## J  META-REGRESSION FOR ICL EFFICACY WITH NUMBER OF DEMONSTRATIONS

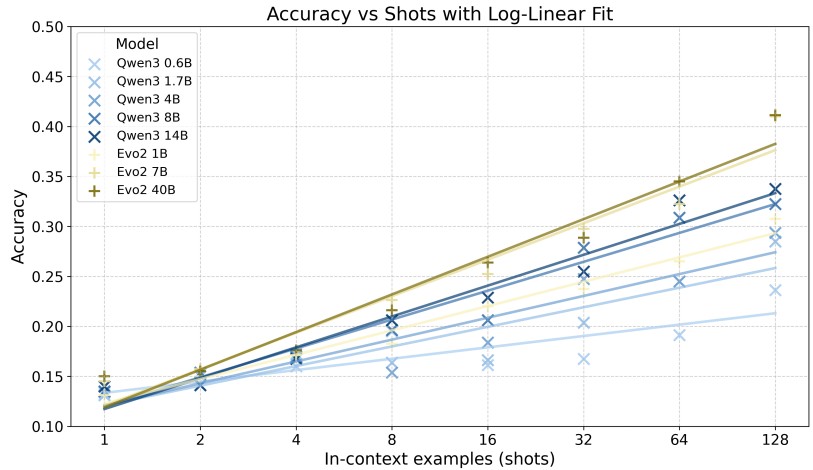

(a) Few-shot performance of Qwen3 and Evo2 models. All models show consistent linear improvement with respect to $\log(\text{shots})$. In contrast, no such improvement occurs for the naive baseline.

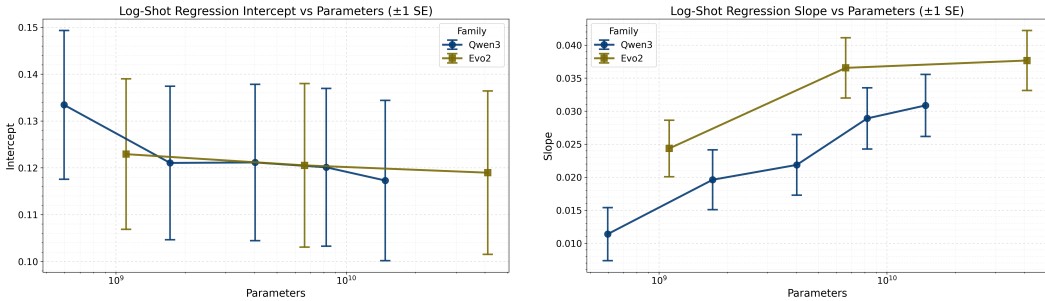

(b) Baseline accuracy decreases slightly with scale as the model gains more parameters for both Evo2 and Qwen3.

(c) ICL rate vs. model size: sharp gains up to 4B for Qwen3; mild boost from Evo2 1B to 7B; both plateau after 4–7B.

Figure 5: Few-shot behavior and scaling trends across Qwen3 and Evo2.

We perform linear regressions to predict accuracy from shot count with each model. For each model $M$, fit the linear regression: $\hat{P}(M, n) = \alpha_0(M) + \alpha_1(M)\log(n) + \varepsilon$. The raw regressions are shown in Fig. 5a. Predictably, all $\alpha_1$ are positive as all models are capable of learning in-context.

We can interpret $\alpha_0$ as representing the model's base accuracy at the task, what it would logically achieve with only one shot to identify the task. We can then interpret $\alpha_1$ as the model's ICL efficacy: the speed at which it adapts to the task being presented and at which its accuracy improves. Analyzing how these values change across parameter values reveals insights into the ICL abilities of both the Qwen3 and Evo2 models.

First, we analyze how $\alpha_0$ changes in each model family – this analysis can be seen in Fig. 5b. Both Evo2's and Qwen3's initial $\alpha_0$ remains essentially constant model-to-model, indicating that all models have similar levels of few-shot baseline performance. Notably, Evo2 and Qwen3 have essentially identical intercepts at around 0.12. This implies that despite drastically different training data, the overall amount of prior knowledge the models have coming into this task is roughly similar. This rules out Qwen simply having 'less experience' with this sort of task.

If one looks at $\alpha_1$ – ICL efficacy – in Fig. 5c, a dramatically different picture is painted. Here, both Qwen3 and Evo2 follow similar patterns with a significant difference. Qwen3's ICL efficacy increases monotonically with respect to parameters, more than doubling from the 0.6B to the 14B. Evo2 follows suit (albeit less dramatically), with ICL efficacy monotonically increasing from the 1B to the 40B.

In absolute terms, however, Evo2, when parameter-matched, adapts in-context faster than Qwen3 does. Evo2 40B outperforms Qwen3 14B significantly, and it takes until Qwen3 8B to exceed the ICL ability of Evo2 1B.

Taken together, this data suggests that Qwen3 and Evo2 have similar amounts of pretraining exposure to be able to solve these tasks, and that Evo2 simply has better overall ICL capability (in this regime) – even though Qwen's ICL ability increases more rapidly with respect to parameters.

## K    DEMONSTRATION OF EVO2'S ICL ABILITY ON A GENOMIC TASK

We demonstrate that Evo2 7B can perform few-shot in-context learning (ICL) on `human_nontata_promoters`, a binary genomic classification task in which the model must predict whether a 251 nucleotide-long human DNA sequence is a promoter. To make the task non-trivial, sequences in the positive class are restricted to non-TATA promoters, removing the "TATA box", a pattern that serves as an obvious, exploitable marker (Umarov & Solovyev, 2017; Grešová et al., 2023).

For each test query, we construct a few-shot prompt by concatenating balanced examples from the training set (equal numbers of promoters and non-promoters), followed by a held-out test sequence. Each training example is immediately followed by an encoded label. The label consists of a 24 nucleotide buffer (a single nucleotide repeated 24 times) and then a 24 nucleotide label run (another nucleotide repeated 24 times). The buffer nucleotide and the two label nucleotides (for class 0 vs. class 1) are re-sampled uniformly at random without replacement from $\{A, C, G, T\}$ for every trial, so the mapping changes across evaluations. This prevents any nucleotide-specific biasing effects.

To predict the test label, we score two candidate prompts - one where the test sequence is followed by the class-0 label and one followed by the class-1 label - and choose the label with lower perplexity on the label run (i.e., computed only over the final 24 label nucleotides, excluding the buffer). This lets us measure whether Evo2 can infer the label mapping from the few-shot context and apply it to the query sequence.

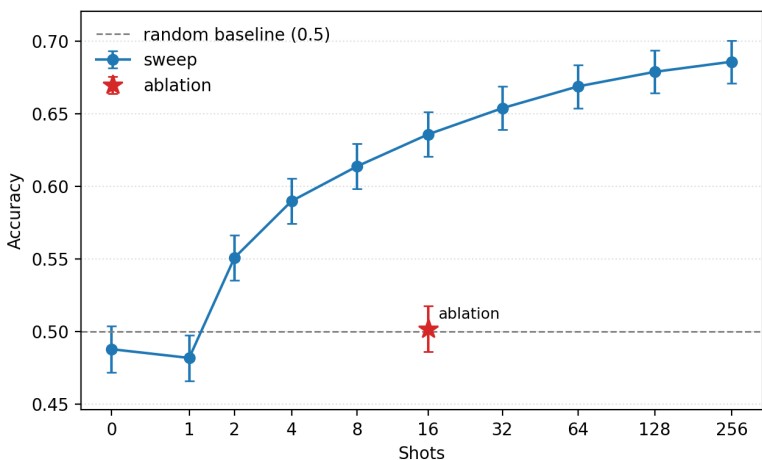

Figure 6: Performance of Evo2 7B on human_nontata_promoters at varying shot counts. Error bars are $\pm$ one standard error. Evo2 7B's performance shows clear monotonic improvements with respect to shot count.

In Figure 6, we observe monotonic improvement with respect to shot count, with accuracy rising from below random baseline at 1-shot (48.2%) to 63.6% at 16-shot. These gains further rise to 68.6% after 256 shots. A saturated power law achieves an $R^2$ of 0.998 in the 2-256 shot regime, which is characteristic of few-shot ICL. Furthermore, we show via ablation that the improvement is due to correctly-labeled examples presented in context—randomly permuting the labels in the 16-shot regime drops performance from 63.6% back to random chance (50.2%).

## L  ANALYSIS OF ICL WITH PROGRAM SYNTHESIS TASKS: BIT-DIVERSITY

BitLoad captures a measure of theoretical information bottleneck required to make predictions in program synthesis tasks. However, this does not always correlate with what models find easy to do. In Fig. 8 and Fig. 9, we plot the accuracy of Evo and Qwen models with respect to BitLoad. We notice that few high BitLoad tasks have high accuracy compared to all medium BitLoad tasks, illustrating that some high BitLoad tasks can be easy for these models to solve, probably due to the nature of patterns found during pre-training.

To further analyze the nature of ICL exhibited through these tasks, we define BitDiversity as *the number of minority bits in the output string*. In Fig. 10 and Fig. 11, we plot the accuracy with respect to BitDiversity. These plots try to estimate the effect of entropy in the output on model performance, and we see a more expected trend: models tend to perform better on low-entropy outputs, regardless of BitLoad. However, bin-wise performance trends are always increasing with shots, supporting the central hypothesis that ICL increases with increasing number of demos in these models.

**Prevalence of $0$-BitDiversity outputs.**  Looking at the expected and predicted outputs of trials from our tasks (Fig. 12), we made a few interesting observations about 0-BitDiversity (BD) outputs.

- Around 25% of true targets are 0-BD. This is a significantly high number as we only have 2 0-BD bit strings in all possible bit strings of any length $K$. It implies that many of our tasks create low BitDiversity outputs on random inputs, i.e., 0-BD outputs.
- Models tend to produce a lot of 0-BD outputs, much higher than the number of 0-BD true targets in the low-shot regime. But this number quickly drops to the expected number with higher shots.
- A large portion of the baseline (1-shot) performance can be explained by this prevalence of 0-BD outputs, but with more shots we get stronger evidence of ICL with increasing correctly predicted non 0-BD cases.

**Understandable Mistakes**. A potential confound is what we will call "understandable mistakes". These mistakes occur when the model outputs an incorrect answer that would be correct for some other programs given the few-shot context. Formally, a model's output $y$ (see 2.2 for notation):

$$y = M\Big(e_1 \to f(e_1), e_2 \to f(e_2), \dots, e_N \to f(e_N), x\Big)$$

is an **understandable mistake** if $y \neq f(x)$ but there exists $f_2 \in F$ such that $\forall e_i \quad f(e_i) = f_2(e_i)$ and $y = f_2(x)$.

These understandable mistakes are an alarming confound at low shot counts, but their effect vanishes by $N = 16$. They occur most in tasks with low BitDiversity, which can often be confused with each other. Figure 7 below shows how understandable mistakes in Qwen3-4B's inferences decay exponentially as the number of shots increases.

**Noise due to Monte Carlo Trials**. We use $m = 8$ for our per-function Monte Carlo trials when we compute accuracy. This has little aggregate impact when comparing model performance across the entire suite, but drastically reduces the significance of results when comparing models and trends at the task-level. We only have eight discrete bands of accuracy at which we can estimate a model's per-task performance – which reduces the expressiveness of regression. Worse, this can lead to models getting a lucky prompt or two with a 0-BD output, which raises performance to 12.5% or 25% without the model truly understanding the task. Addressing these confounds would simply require increasing $m$ by an order of magnitude or so. Alternatively, tasks of interest could be identified and $m$ selectively increased for those tasks to enable more nuanced analysis.

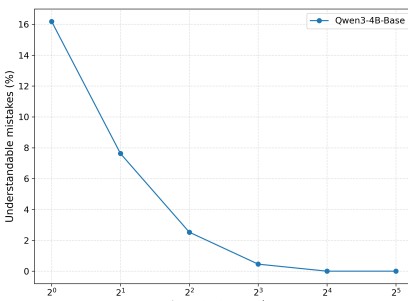

Figure 7: Qwen3-4B's rate of understandable mistakes with respect to the number of shots. Despite starting at 16% in the one-shot regime, they fall to less than 1% by 8 shots and vanish entirely at 32 shots. This underscores how understandable mistakes are only a confound at very low shot counts and can essentially be ignored past 4 shots.

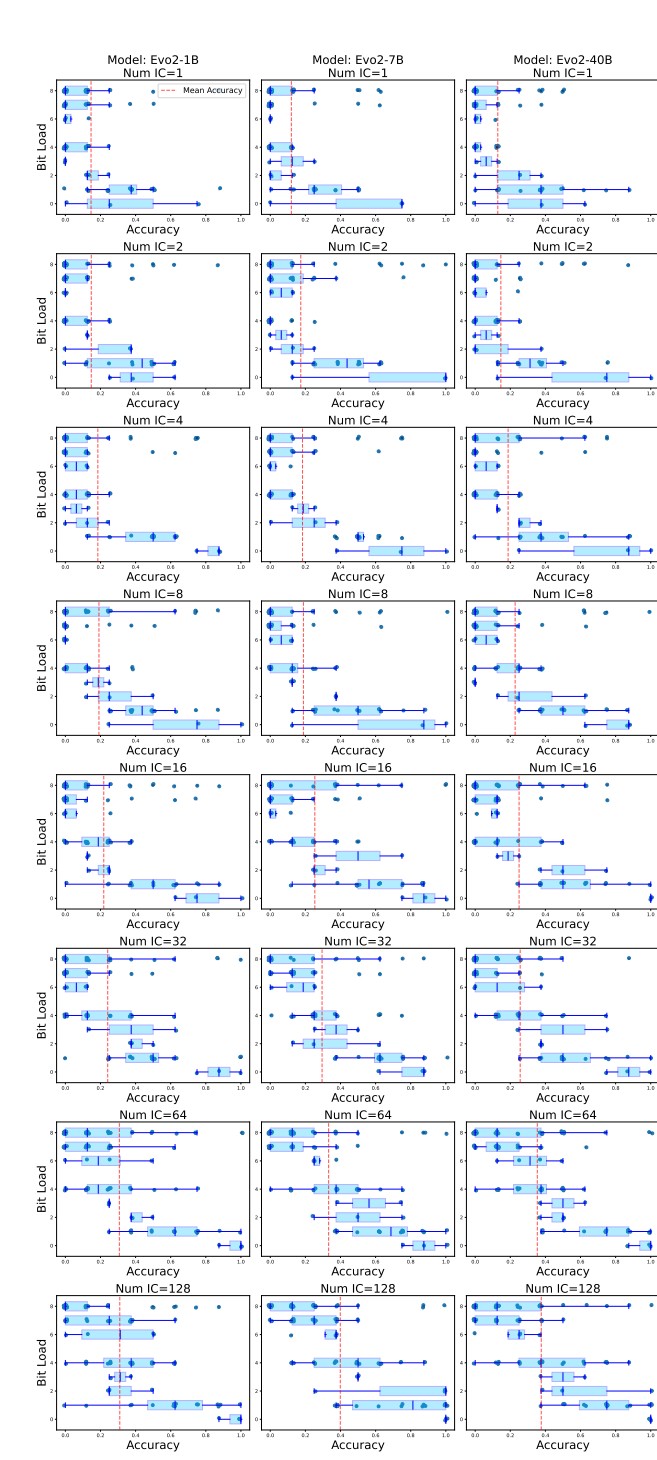

Figure 8: BitLoad vs Accuracy for all Evo models at all ICL shot-numbers. The per-BitLoad distribution of model performance at each shot level shows an uneven affinity of models for solving tasks at different BitLoads. However, all trends support our overall hypotheses.

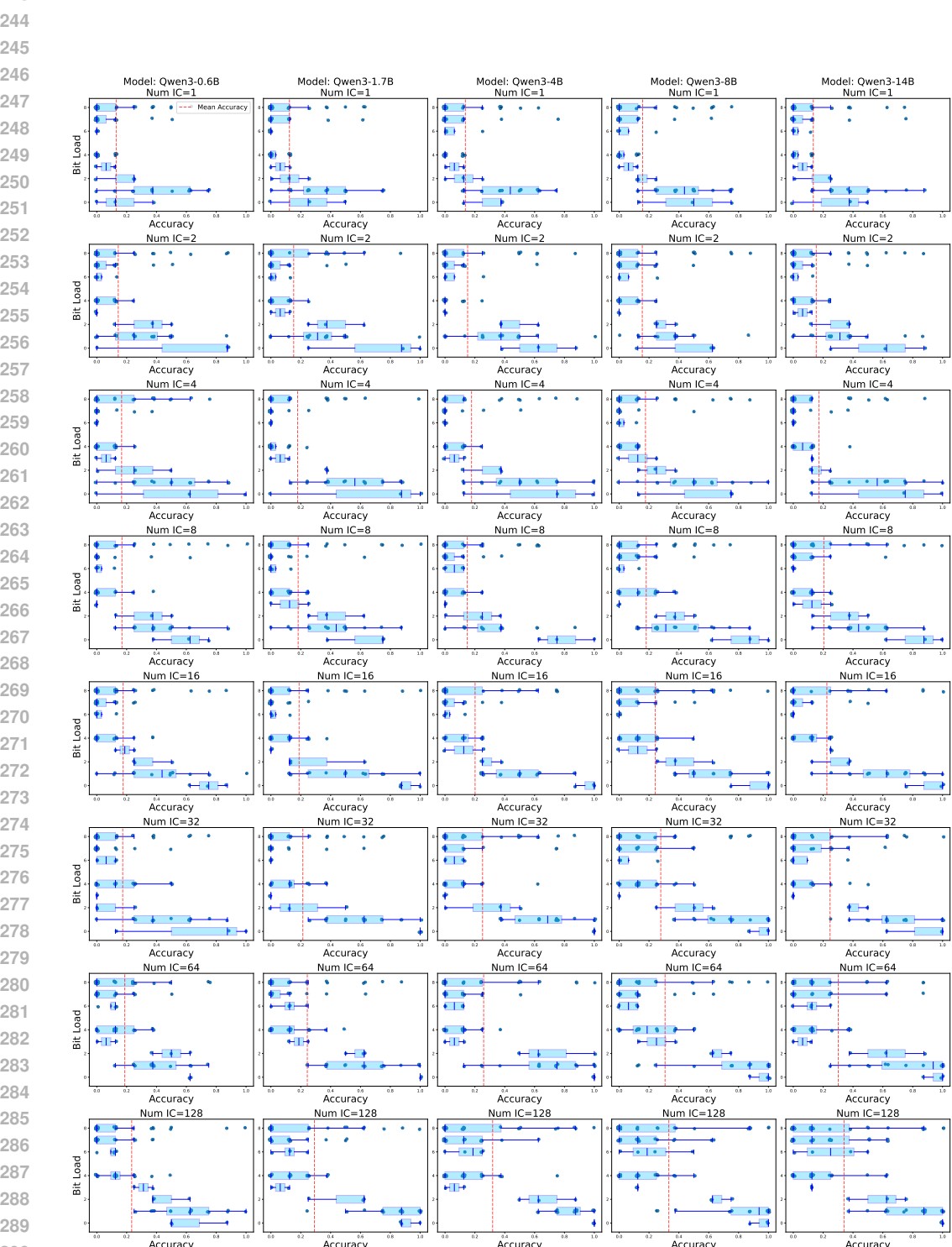

Figure 9: BitLoad vs Accuracy for all Qwen models at all ICL shot-numbers.

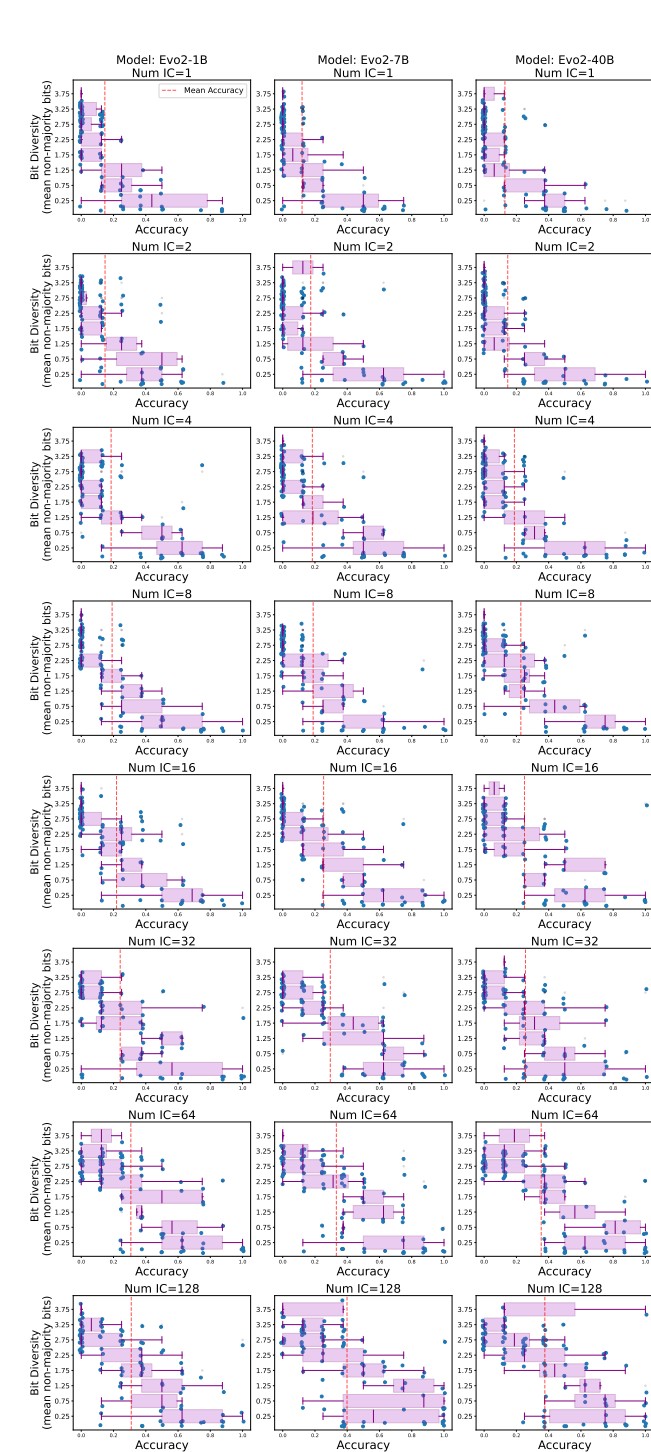

Figure 10: BitDiversity vs Accuracy for all Evo models at all ICL shot-numbers. Model performance follows a more natural pattern of increasing performance with decreasing output entropy. This highlights that it is difficult for models to decipher ICL patterns for high entropy outputs.

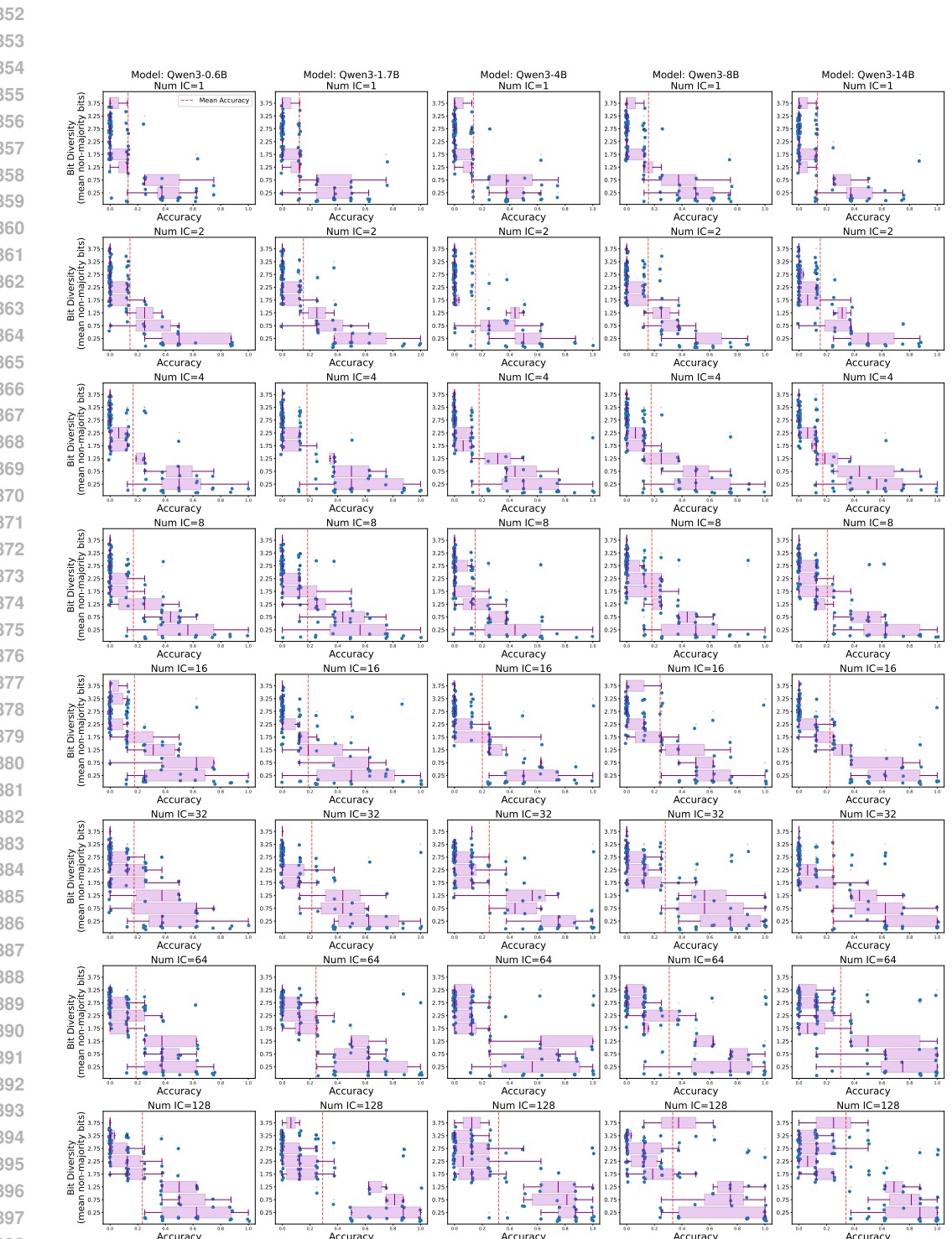

Figure 11: BitDiversity vs Accuracy for all Qwen models at all ICL shot-numbers.

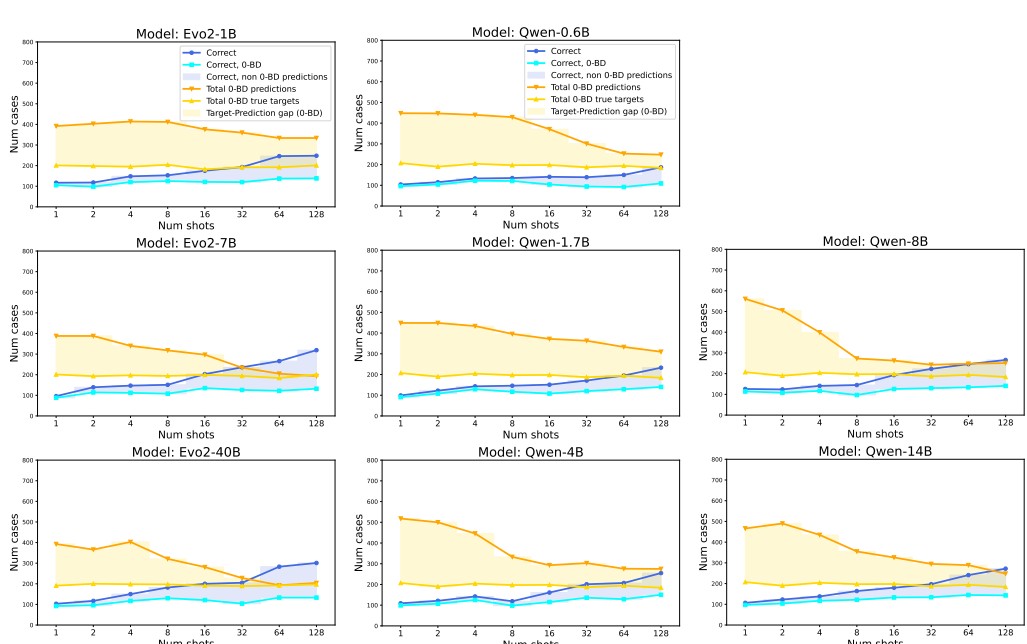

Figure 12: Enumerating cases of 0 BitDiversity. For our defined tasks, around 25% of the true targets have 0 BitDiversity4 (BD). With low-shots, models tend to produce a large number of 0-BD predictions. But this number decreases significantly with increasing shots and tends to match the actual prior value. Similarly, a majority of the baseline (1-shot) performance of models can be explained through 0-BD outputs. With a higher number of shots, the model starts learning higher entropy patterns and presents stronger evidence of ICL.

