# OpenReview forum: "Genomic Next-Token Predictors are In-Context Learners"
_ICLR.cc/2026/Workshop/Sci4DL — Sci4DL 2026_

### Official Review · Reviewer_MGZi · 2026-02-26

**Fit:** 3
**Significance:** 2
**Confidence:** 2

**Summary:**

This paper investigates whether in-context learning (ICL), typically studied in large language models, can also emerge in a genomic next-token prediction model (Evo2). To evaluate this, the authors design a suite of symbolic reasoning tasks that can be encoded both in natural language and in genomic sequence form. By testing Evo2 on these tasks, the paper provides empirical evidence that the model exhibits ICL-like behaviour despite being trained on nucleotide sequences rather than language. The results suggest that in-context learning may arise from the next-token prediction objective itself, rather than being specific to linguistic training data.

**Strengths:**

**Clear, focused empirical question.*** The paper addresses a concrete and well-motivated question: whether ICL depends on language-specific data or can emerge in other sequence domains. This framing is compelling and relevant to broader discussions about the origins of in-context learning, which is interesting to the Sci4DL community (see also third point).

**Creative benchmark design.** The construction of symbolic reasoning tasks that can be encoded in both natural language and genomic sequences is conceptually elegant. This cross-domain formulation provides a useful testbed for probing sequence models in a modality-agnostic way.

**Broader conceptual implications.** The findings are consistent with the hypothesis that ICL may be a consequence of the next-token prediction objective and transformer architecture, rather than something uniquely tied to linguistic structure. This has implications for how we think about generalisation in sequence models.

**Suggestions:**

**Clarify the hypothesis and the scope of the causal claim.** While the results are consistent with the idea that ICL emerges from the optimisation objective (next-token prediction in this case), they do not fully isolate the cause. Architectural inductive biases or token distribution properties could also play a role. It would strengthen the paper to more explicitly discuss these alternative explanations and position the findings as evidence consistent with an explicitly stated hypothesis.

**Control and analyse task complexity.** The symbolic reasoning tasks would benefit from a more systematic treatment of complexity. For example, can the number of reasoning steps, compositional depth, or dependency length be controlled? Studying performance as a function of task complexity could help distinguish shallow pattern matching from more compositional reasoning and would provide a clearer characterisation of the model’s capabilities.

**Investigate potential overlap with the pretraining objective.** Since genomic sequences operate over a small token alphabet, it would be helpful to rule out the possibility that performance on the synthetic tasks is partially driven by potential overlaps with the original training distribution. Possible ablations could include:

-Shuffled or structure-controlled variants of the tasks (see second point);

-Comparisons to models trained on randomised genomic data.

Such analyses would strengthen the interpretation that the observed behaviour reflects genuine in-context learning rather than surface-level statistical alignment.

**Formalise the task encoding framework.** The paper could be strengthened by more explicitly abstracting the task-to-sequence mapping. If the symbolic tasks can be systematically converted into fixed-vocabulary token sequences independent of modality, this would clarify the generality of the approach and potentially enable reuse across other sequence domains.

Overall, the paper presents an interesting empirical observation with promising implications. With additional analysis and stronger controls, it could offer a more definitive contribution to the understanding of in-context learning beyond language models.

---

### Official Review · Reviewer_yxpG · 2026-02-27

**Fit:** 2
**Significance:** 2
**Confidence:** 3

**Summary:**

This paper investigates if ICL emerges in non-linguistic domains by testing on theEvo2 genomic model. The authors use map bitstring reasoning tasks to both genomic and language compatible formats and show that Evo2, just like the Qwen3 language models exhibits clear log-linear accuracy improvements with number of in-context shots. These findings suggest that ICL is not unique to llms but more generally observable in pattern-rich sequential data.

**Strengths:**

- Novel research question: First demonstration of organically emergent ICL in a non-linguistic (genomic) domain.
- Clean cross-domain design: Bitstring tasks encodable in both nucleotide and digit alphabets enable fair apples-to-apples comparison across modalities.

**Suggestions:**

- "Intelligence at the Edge of Chaos" (Zhang et al., 2024) has already showed ICL-like capabilities emerge from training on complex non-linguistic sequences (cellular automata), so framing this as the "first evidence" might be a stretch.

---

### Meta-Review · Area_Chair_gN3a · 2026-02-28

**Recommendation:** Accept

**Metareview:**

This paper demonstrates that the Evo2 model, trained on genomic data, exhibits ICL capabilities. Both reviewers noted the originality and significance of the paper. I recommend acceptance.

---

### Decision · Program_Chairs · 2026-03-02

Accept